# Regulation of Poly-E Motif Flexibility by pH, Ca^2+^ and the PPAK Motif

**DOI:** 10.3390/ijms23094779

**Published:** 2022-04-26

**Authors:** Sudarshi Premawardhana Dassanayake Mudiyanselage, Matthew J. Gage

**Affiliations:** 1Department of Chemistry, University of Massachusetts Lowell, Lowell, MA 01854, USA; sudarshi_dassanayake@student.uml.edu; 2UMass Movement Center (UMOVE), University of Massachusetts Lowell, Lowell, MA 01854, USA

**Keywords:** PEVK, poly-E, intrinsically disordered protein, pH, titin

## Abstract

The disordered PEVK region of titin contains two main structural motifs: PPAK and poly-E. The distribution of these motifs in the PEVK region contributes to the elastic properties of this region, but the specific mechanism of how these motifs work together remains unclear. Previous work from our lab has demonstrated that 28-amino acid peptides of the poly-E motif are sensitive to shifts in pH, becoming more flexible as the pH decreases. We extend this work to longer poly-E constructs, including constructs containing PPAK motifs. Our results demonstrate that longer poly-E motifs have a much larger range of pH sensitivity and that the inclusion of the PPAK motif reduces this sensitivity. We also demonstrate that binding calcium can increase the conformational flexibility of the poly-E motif, though the PPAK motif can block this calcium-dependent change. The data presented here suggest a model where PPAK and calcium can alter the stiffness of the poly-E motif by modulating the degree of charge repulsion in the glutamate clusters.

## 1. Introduction

Titin is a giant elastic protein that is the third major filament in the sarcomere, and while it has been studied for over four decades, many questions remain about titin function [1,2,3]. The primary elastic region of titin is found in the I-band of the sarcomere and is comprised of two main types of structural domains: repeated immunoglobulin (Ig) domains and the PEVK region [4,5,6,7,8]. The role of the Ig domains in elasticity continues to be debated, but it is well-established that the PEVK region becomes elongated during moderate to long stretching of striated muscle, resulting in a steep increase in passive tension [9,10]. Early studies of PEVK’s function modeled this elongation as an entropic spring [11]. However, more recent titin-based myofibrillar studies have identified an ionic strength dependency of PEVK and bending rigidity of PEVK is reduced under high calcium levels [10,12]. This suggests that PEVK exhibits characteristics of an enthalpic spring as well and that elasticity of PEVK is not purely entropic [10].

The PEVK region has a disordered nature, with 75% of its amino acid content composed of proline (P), glutamic acid (E), valine (V), and lysine (K). The sequence of the PEVK region can be divided into two motifs, commonly called the PPAK motif and the poly-E motif [13]. PPAK motifs generally start with the amino acids sequence P-P-A-K, contain 26–28 residues, and have an approximately 19% lysine content [14]. Interspersed among the PPAK motifs are glutamate-rich regions, called poly-E motifs, composed of clusters of 3–4 glutamates separated by between 1–3 hydrophobic residues. These poly-E motifs have a high negative charge density due to their high glutamic acid (~45%) content [14].

The extension of the PEVK region is hierarchical, occurring in three steps. The C-terminal sub-fragment is the most compliant region, followed by the middle section of the PEVK and the N-terminal sub-fragment is the stiffest region [15]. The hierarchical nature of PEVK extension is correlated with specific sequence features. The C-terminal sub-fragment has a lower composition of glutamate residues, while the N-terminal sub-fragment of PEVK has a much higher glutamate density per exon [14]. This results in the N-terminal sub-fragment having a more rigid structure than the other two sub-fragments due to more electrostatic stiffening [15]. While some properties of the PEVK are understood, there are still enough unknown features regarding how various sequence motifs interact with each other in response to mechanical forces and solution conditions that it remains difficult to model the mechanism of how this region functions during eccentric stretching.

Based on the density and clustering of negative charges in poly-E motifs, it was hypothesized that these motifs might be sensitive to fluctuations in ionic strength and pH within the sarcomere during contraction. Our lab has tested this hypothesis using 28 amino acid peptides based on PPAK and poly-E sequences from human titin [16]. These studies demonstrated that the conformation of the PPAK motif peptide remains constant over the tested pH range [16]. In contrast, the poly-E-derived peptide has an extended conformation at physiological pH but becomes more flexible as the pH decreases [16]. These results are consistent with our model of the peptide having extensive charge repulsion at pH 7, which is reduced as glutamates become protonated at lower pH. Interestingly, this conformational shift begins to occur at a higher pH than predicted by the pK_a_ of free glutamate, suggesting that the local environment is shifting the pK_a_ toward physiological pH [16].

As seen in several other studies on the PEVK region [13], our initial studies were conducted with peptides that represented either the PPAK or poly-E motif but not a combination of both sequences. We hypothesized that the PPAK motif could modulate pH-dependent conformational changes in poly-E regions, altering the conformational flexibility of this region. We tested this hypothesis using three recombinantly expressed constructs that consisted of either three consecutive PPAK motifs (PEVK1), a poly-E motif sequence (PEVK2), or both the poly-E motif and PPAK motifs in the same construct (PEVK3). Our results demonstrated that both poly-E-containing constructs (PEVK2 and PEVK3) undergo conformational shifts as a function of pH and shifts in ionic strength, though the inclusion of the PPAK motif sequence modulates the conformational shifts. The PEVK2 construct also exhibited a Ca^2+^-dependent conformational change at physiological pH that was reduced at lower pHs. Taken together, these results support a mechanism where poly-E motifs can undergo conformational shifts due to changes in pH or ionic strength, but these conformational changes can be modulated through interactions with PPAK motifs.

## 2. Results

### 2.1. Long Poly-E and Poly-E-PPAK Construct Exhibit pH Dependent Conformational Changes

The PEVK region of titin elongates during moderate stretching of striated muscle, resulting in an increase in passive tension [9] but how this is achieved is still unknown. Since this is a disordered region, many of the principles associated with the mechanical unfolding of globular proteins are not conserved. The glutamate-rich poly-E motifs have been shown to contribute significantly to PEVK’s function, and work by our lab has shown that fluctuations in pH alter the conformational flexibility of these regions [12,16]. This change in flexibility is driven by changes in the ionization of the glutamate residues in these regions, but little is known about how PPAK motifs might modulate these conformational changes.

To explore this relationship, we developed a construct that contained the longest poly-E motif sequence along with two PPAK motifs on the N-terminal side of the poly-E motif and one on the C-terminal side of the motif (PEVK3, Figure 1). We used constructs that contained just the three PPAK motif sequences with the poly-E sequence deleted (PEVK1) or the isolated poly-E motif sequence (PEVK2) as controls in these experiments. Based on our previous results [16], we predicted that our control constructs (PEVK1 and PEVK2) would show similar behavior as observed in our peptide studies [16]. However, we postulated that the positively charged PPAK motifs in PEVK3 might interact with some of the negatively-charged glutamate clusters and alter the response to pH shifts. 

We tested this hypothesis using a combination of circular dichroism (CD) and size exclusion chromatography (SEC). In both cases, the samples were prepared in a 20 mM potassium phosphate buffer with 150 mM KCl, and the pH was adjusted to the desired level. Consistent with our previous studies, the PEVK1 construct, which contains only PPAK sequences, did not exhibit any significant changes in conformation in either the SEC (Figure 2a) or CD (Figure 2b). Changes were observed in the retention time (Figure 2a) and the 200 nm minimum (Figure 2b) in the PEVK2 and PEVK3 constructs, as predicted based on their glutamate content.

While both constructs exhibited the predicted response to shifts in pH, the PEVK2 and PEVK3 constructs did not exhibit identical behavior, suggesting that the PPAK motif modulates the poly-E motif’s response. Most notably, the addition of the PPAK domains reduced the pH sensitivity of the constructs in both experiments. The impact was more pronounced in the CD data, where the minima measured for the PEVK3 constructs were more like the PEVK1 construct than the PEVK2 construct. Our previous peptide-based study suggested that the poly-E sequence has a coil-like structure above pH 6, shifting toward the pre-molten globule state at pHs below six. A similar shift is observed with the PEVK2 construct, while the PEVK3 construct has less of a coil-like conformation at pHs above six, suggesting that the PPAK sequences are potentially interacting with the poly-E regions to partially disrupt the coil-like conformation observed with the pure poly-E sequence.

The impact of pH on the hydrodynamic radius of the constructs as measured by SEC was similar but did not exactly mirror the effects observed by CD. With both sets of measurements, there was a plateau at pH 3–4 where the glutamates are protonated, and therefore conformational changes would be expected over this pH range. In both sets of measurements, there was a conformational transition that occurred between pH 4 and 8, though the nature of this transition varied between the two techniques. With the SEC measurements, the change in retention time between each pH step was equivalent for the PEVK2 and PEVK3 constructs, resulting in a similar slope for the transition.

In contrast to the SEC measurements, the transition slope for the PEVK3 construct was significantly shallower than the PEVK2 construct in the CD measurements. This suggests that the poly-E sequences in the two constructs do not undergo equivalent conformational transitions as a function of pH. The PEVK2 construct undergoes a relatively steep transition that mirrors the retention time change observed by SEC. The introduction of the PPAK motif sequences reduces the pH sensitivity of the PEVK3 construct in the CD data. This observed difference is likely to be a secondary structure effect rather than a change in the hydrodynamic radius since the transitions observed in the SEC data for PEVK2 and PEVK3 between pH 4 and 8 have similar slopes

The second interesting feature of the SEC data is the similarity in the retention times of the PEVK2 and PEVK3 constructs at pH 3 and 4. This suggests that the PPAK motif collapses onto the poly-E sequence in the PEVK3 construct to give a similar Stokes radius, even though there is an ~9 kDa difference in molecular weight. Once the pH increases, there is a shift between the PEVK2 and PEVK3 constructs, suggesting that deprotonation of the poly-E introduces some charge repulsion that results in a less compact form of the protein that is distinguished between the different molecular masses.

The CD and SEC results presented in this paper show a similar conformational transition in poly-E containing polypeptides as we observed in our previously published peptide study [16]. However, the shorter, 28-amino acid peptides used in our previous study exhibited no conformational changes between pH 8 and 6, followed by a gradual change in conformational flexibility between pH 6 and 4 [16]. In contrast, the longer constructs used in this study exhibited a plateau between pH 3 and 4, and then a gradual shift in conformational flexibility was observed between pH 4 and 8. Our peptide studies did not extend to pH 3; therefore, a similar low pH plateau that is comparable to the longer constructs might be observed if those studies were extended. However, the lack of a plateau in the conformational transition between pH 6 and 8, as observed in the peptide studies, suggests that the increased size of this region results in a broader range of pH sensitivity, supporting the hypothesis that the poly-E sequences might play a role in modulating elasticity as a function of pH.

### 2.2. Calcium Ions Alter the Disordered Structure of Isolated Poly-E Motifs

Calcium is a well-established regulator of muscle function, and at least one calcium-binding site has been experimentally confirmed to exist in Exon 129, one of the poly-E motif exons [12]. This exon has been shown to have a shorter persistence length when calcium is bound [12], suggesting that calcium could modulate the stiffness of PEVK by neutralizing charge repulsion in poly-E motifs. Given that calcium levels change in response to activation, it is a reasonable hypothesis that the binding of calcium to poly-E regions could be a mechanism for regulating compliance of the PEVK region in the activated state [17,18]. If this hypothesis is correct, we predicted that poly-E motif-containing constructs would exhibit a calcium-dependent increase in conformational flexibility.

We tested this hypothesis by measuring the conformation of our three constructs between pCa 10 to pCa 3 at pH 7.5. As predicted, the PEVK2 construct exhibited a conformational shift as calcium concentrations increased (Figure 3a). This shift was linear over the tested concentrations, which covers the calcium concentration range associated with muscle activation. To confirm that this conformational change was due to calcium binding to the glutamate residues, we repeated this experiment with the PEVK2 construct at pH 3.5 so that the glutamates were in an uncharged state. No conformational shifts were observed under these conditions, supporting our model that the calcium is binding to the negatively charged glutamate clusters (Figure 3b).

Similar experiments were conducted on the PEVK1 and PEVK3 constructs, and no conformational shifts were observed for either of these constructs. This was expected for the PEVK1 construct since there are no glutamate clusters in this sequence to provide a calcium-binding site. Surprisingly, the PEVK3 construct did not exhibit any conformational changes between pCa 10 and pCa 4. The PEVK2 and PEVK3 constructs contain the same poly-E sequence; therefore, we had predicted that similar conformational changes would be exhibited in both constructs. The lack of a conformational change in the PEVK3 construct suggests that the PPAK motifs interact with the glutamate clusters on the poly-E motif to occlude the calcium-binding sites, either through direct interaction between the PPAK and the glutamate clusters or by indirectly blocking the calcium-binding site.

We also wanted to determine whether this was a calcium-specific effect or a divalent metal effect. This was tested by repeating these experiments using the PEVK2 and PEVK3 constructs in the presence of Mg^2+^ instead of Ca^2+^. Unlike what was observed with Ca^2+^, no significant conformational shifts were observed for either construct over the range of Mg^2+^ tested (Figure 4). This suggests that the conformational shifts observed when Ca^2+^ was present are specific to this metal and not a general divalent metal effect. We hypothesize that the glutamate clusters can form a binding pocket that is ideally sized so that Ca^2+^ can stabilize this pocket, but Mg^2+^, being smaller, cannot provide the same stabilization.

### 2.3. PEVK2 and PEVK3 Undergo Sodium and Potassium Dependent Structural Changes

It is a well-accepted principle that ionic strength can mediate protein-protein interactions and the flexibility of disordered proteins [19,20,21]. The results presented earlier for Ca^2+^ and Mg^2+^ focused on how the potential binding of these divalent metals to the constructs might modulate conformational flexibility. To assess the impact of ionic strength shifts on the conformational flexibility of the poly-E motif-containing constructs, we repeated the CD measurements in the presence of a range of Na^+^ and K^+^ concentrations. Our working hypothesis was that increased ionic strength would provide electrostatic shielding to the negatively charged glutamates, helping to reduce charge repulsion.

Both constructs showed shifts to more negative CD signals at 200 nm with increasing ionic strength, consistent with our hypothesis (Figure 5) that increased ionic strength will increase conformational flexibility. A similar response to changes in conformation was observed with both constructs over the range of K^+^ concentrations tested (Figure 5a). There is an immediate shift in conformational flexibility between 0 mM KCl and 50 mM KCl, and then the conformational flexibility was consistent between 50 mM KCl and 250 mM KCl, which spans the physiological concentration of K^+^. There is an additional shift in conformation at concentrations beyond the normal physiological range, suggesting that a large K^+^ spike would induce increased flexibility in the PEVK region of titin.

The PEVK2 construct, which only has the poly-E sequence, follows a similar pattern of conformational flexibility shifts in the presence of Na^+^ as was observed with K^+^ (Figure 5b). The plateau in conformational shifts occurs between 100 mM and 250 mM, indicating that it takes more Na^+^ than K^+^ to establish a stable conformation. In contrast, there is no significant change in conformational flexibility in the PEVK3 construct until after 100 mM NaCl, and then there is a linear increase in conformational flexibility all the way to 500 mM NaCl, well above normal physiological concentrations. This suggests that the PPAK motif modulates the response of the poly-E motifs to the changes in Na^+^ concentration and that this might represent an additional mechanism for tuning PEVK stiffness.

Glutamate side chains are deprotonated at pH 7.5, and, in the absence of these alkaline metal ions, they are presumably experiencing repulsions between each other, which results in electrostatic stiffening. As sodium or potassium ion concentrations increase, these glutamates will become electrostatically screened. This phenomenon will reduce repulsions between glutamates, which will gradually alter the electrostatically stiffened conformations of PEVK2 and PEVK3. This hypothesis was further supported by size exclusion data of PEVK2 and PEVK3 in the presence of potassium. Both PEVK2 and PEVK3 proteins exhibited longer retention times as potassium concentration was increased (data not shown). In theory, longer retention times represent smaller hydrodynamic sizes of the molecules. As the electrostatic stiffening is reduced due to the shielding of glutamates by potassium, both proteins will gradually gain collapsed conformations which agree with the longer observed retention times at higher potassium levels. As these transitions take place outside the normal physiological range, it is not likely that they are significant.

## 3. Discussion

The PEVK region of titin is a highly extensible, disordered region, contributing to titin’s role in the development of passive tension [10,22]. While this region has an important role in passive tension, many underlying principles governing this process remain elusive. One interesting feature likely related to passive tension development is the heterogeneity of the sequence of the poly-E motifs and their distribution throughout the PEVK region. The molecular-level understanding of how PEVK’s structure dictates its function is still poorly understood. Our previous work and the work of others have shown that poly-E regions are more susceptible to pH and ionic fluctuations in the sarcomere [9,12,16]. The majority of these studies have been conducted with constructs isolating either the PPAK or poly-E sequences, and we sought to determine how the presence of PPAK motifs would impact the conformational changes that have been observed in poly-E motifs under varying conditions.

We expressed and purified proteins that represented three different types of sequences for this study. One construct contained three PPAK motifs (PEVK1), the second consisted of the longest poly-E motif (PEVK2), and the third construct was the poly-E motif with two PPAKs on the N-terminal side of the poly-E motif and one PPAK on the C-terminal side of the construct (PEVK3). Using a combination of CD and HPLC, we demonstrated that PEVK2 and PEVK3 both exhibit pH-dependent conformational changes over a prolonged pH range (pH 4 and pH 8) rather than the sigmoidal transition observed for the deprotonation of a weak acid. This result is consistent with our previous studies using shorter poly-E sequences [16]. The most notable difference with the longer constructs is the lack of a plateau at physiological pH, suggesting that the stiffness of poly-E motif sequences can be modulated by the pH shifts that occur during prolonged muscle use. The presence of the PPAK motif does modulate the degree of the conformational change in the poly-E motif since PEVK3 does not exhibit as significant a structural shift as PEVK2 (Figure 1), suggesting that there are layers to the response of titin to pH shifts.

Our results also demonstrated that physiological shifts in Ca^2+^ can induce shifts in the conformational flexibility of poly-E motifs, though the presence of PPAK motifs appears to compete for the calcium-binding site. This modulation of flexibility does appear to be due to a specific binding event between Ca^2+^ and the glutamate clusters, as a similar response is not observed with Mg^2+^ or at low pH, where glutamate will not be protonated. The conformational flexibility of the poly-E motif can also be modulated by changes in ionic strength, presumably due to electrostatic shielding. Interestingly, the response of the PEVK3 construct, with both the PPAK and poly-E sequences, is unique for Na^+^. Both constructs show a plateau in conformational change over the normal physiological range for K^+^, and a similar plateau is observed for PEVK2 with Na^+^. These results suggest that fluctuations in Na^+^ concentrations might contribute to the modulation of PEVK stiffness.

### 3.1. Calcium Dependency of Poly-E Regions

It is well-established how the influx of calcium during activation is critical for contraction as it exposes myosin-binding sites on the thin filament [23]. What is often not appreciated is the role of calcium in titin function. Labeit et al. demonstrated that the binding of calcium to a poly-E region induces a conformational change that reduces bending rigidity [12]. This is consistent with our results, as a similar conformational change was observed with our poly-E construct. Both studies were conducted using physiological calcium levels, suggesting that calcium may act as a regulator of PEVK flexibility in metabolically active muscle.

It is not possible to conclusively determine if the observed impact of Ca^2+^ is a non-specific effect or is due to specific binding interactions between Ca^2+^ ions and glutamate clusters. However, the results presented in this manuscript suggest that there are specific binding sites for Ca^2+^ in our construct. A similar effect should be observed with Mg^2+^ if the increased conformational flexibility was due to a divalent metal effect. Similarly, if the conformational shift was due to electrostatic shielding, there should be a greater effect observed for either Na^+^ or K^+^. While circumstantial, these results suggest that specific binding sites for Ca^2+^ exist and that additional studies to quantitate the amount of bound Ca^2+^ and potential binding sites would be warranted.

### 3.2. Impact of PPAK on Calcium Binding to Poly-E Motifs

Previous studies have investigated the potential interactions between PPAK and poly-E motifs and have not found any substantial interactions [13,24]. Unpublished results from our lab using the 28 amino-acid peptides from our previous study also did not reveal any significant interactions between the two motifs. This observation seemed counterintuitive since the PPAK motif has an overall net positive charge, which should be capable of forming ionic interactions with the negatively charged poly-E motif. That is further supported by data suggesting that there is an enthalpic component to PEVK’s elasticity [9]. Our results showing that calcium-dependent conformational flexibility of poly-E sequences is lost in the presence of PPAK motifs suggests that the two sequence motifs do interact when they are linked together, which has significant functional implications.

Under resting conditions or if the muscle is undergoing concentric contractions, titin is compacted since the distance between the Z-disk and the thick filament is shorter than titin. Under these conditions, there is no significant force being applied to the extensible region of titin, and therefore the degree of conformational flexibility will not have any impact on titin’s elasticity. If there are interactions between the PPAK and poly-E motifs, they would be maximized under these conditions since the PEVK region would be in a compacted state. However, as the muscle undergoes eccentric contractions or passive stretching, the force will begin to be applied to the PEVK region, and these interactions will begin to be disrupted based on how far the sarcomere is lengthened. During this elongation and stretching of PEVK, the interplay between the association of poly-E with either PPAK or calcium is likely to be mechanistically significant.

The results of this study suggest that the interaction between poly-E and PPAK motifs help increase the conformation flexibility of the poly-E region, which would reduce the overall stiffness of the PEVK. As the PEVK region is passively stretched, these interactions would start to be disrupted, which would increase the stiffness of the poly-E motif. This model is consistent with experimental results that show increased PEVK stiffness as sarcomere lengths are increased during passive stretch [REF]. During an eccentric contraction, the mechanics associated active contraction, like cross-bridge forces, help maintain sarcomere length, and titin stiffness could be detrimental to achieving the desired stretch. In this situation, as titin is stretched and the PPAK/poly-E interactions are disrupted, the influx of calcium associated with eccentric contraction could help maintain flexibility of the poly-E region, reducing PEVK stiffness and allowing longer sarcomere lengths to be achieved more easily. Force experiments with these constructs would help test this hypothesis, but until those can be conducted, this model provides a potential mechanism explaining why PPAK and calcium would exhibit similar effects on the poly-E motif stiffness.

The observation that PPAK motifs can inhibit calcium binding also provides a potential explanation for another puzzling observation regarding the interaction between titin and calcium. Experimental results have suggested that titin binds 12 calcium ions [25], which seems extremely small given the diversity of domains in titin, its overall size, that two Ig domains are stabilized by calcium [26,27,28], and the potential number of calcium-binding sites in the poly-E regions [12]. The observation that PPAK motifs can occlude calcium-binding sites in poly-E motifs provides a plausible mechanism for why the amount of calcium bound by titin is not thought to be higher. These experiments were conducted under conditions where the PPAK motif would be blocking the majority of poly-E sites, and our data suggest that under physiological conditions, the PPAK/poly-E interaction outcompetes the Ca^2+^/poly-E interaction.

### 3.3. Role of PEVK in Residual Force Enhancement

One of the more poorly understood properties of muscle is Residual Force Enhancement (RFE). RFE is the increase in isometric force following an eccentric contraction relative to the isometric force from a purely isometric contraction. Experimental evidence has linked titin to RFE [29,30,31], but the mechanism of how titin contributes to this effect is still unclear. Currently, the most accepted hypothesis is that a stiffening of the titin protein in active muscle is a contributing factor [32].

There are two hypotheses to explain the increase of titin stiffness in active muscle: increase of titin’s inherent stiffness due to a change in its material property or shortening of titin’s spring length while its material property is unaltered [33]. The increase of titin’s inherent stiffness has been explained by the calcium dependency of titin. Labeit et al. have observed an approximately 20% increase in non-crossbridge force in skinned mouse soleus fibers when activated with calcium (pCa 4) compared to passively stretched fibers [12]. Other studies have shown stiffening of the I27 domain and a poly-E region under physiologically relevant calcium levels [26].

Our results are consistent with calcium influencing the stiffness of titin, though they suggest that calcium is helping to reduce stiffness by decreasing charge repulsion by binding to glutamate clusters. However, our studies are using a single poly-E motif, and in full-length titin, there will be multiple poly-E motifs. It is possible that the observed stiffening of titin in the presence of calcium in other studies might be (1) due to the stabilization of other domains or (2) from calcium acting as a bridging ligand between glutamate clusters from different poly-E motifs. In this model, the titin would be shortened due to the formation of hairpin structures held together by these Ca^2+^-bridge salt bridges. The absence of magnesium-dependent structural changes of poly-E also support our proposed hairpin model, where calcium would bind poly-E motifs at specific glutamate sites bringing together distant subregions of poly-E, resulting in increased stiffness of PEVK. When calcium levels increase in an active muscle (from 100 nM to 1.4 μM), these hairpins will change the material property of titin, increasing its stiffness, potentially contributing to RFE.

## 4. Materials and Methods

All the chemicals used were purchased from standard chemical suppliers such as Fisher Scientific (Hampton, NH, USA). The coding regions were picked from titin exons 142–150 and optimized for expression in *Escherichia coli* by Life Technologies (Carlsbad, CA, USA). The synthesized genes were subcloned into a lab-designed plasmid that secretes expressed proteins into the growth media. The three proteins used in this study were PEVK1, PEVK2, and PEVK3 representing three consecutive PPAK (italic) motifs, a long poly-E (underlined) region, and a combination of PPAK and poly-E, respectively.

PEVK1:
*PPAKVPEVPKKPVPEEKVPVPVPKKVEAPPAKVPEVPKKPVPEKKVPVPAPKKVEAPPAKVPEVPKKLIPEEKKPTPVPKKVEA*
PEVK2:
PPPKVPKKREVPVPVALPQEEEVLFEEEIVPEEEVLPEEEEVLPEEEEVLPEEEEVLPEEEEIPPEEEEVPPEEEYVPEEEEFVPEEEVLPEVKPKVPVPAPVPEIKKKVTEKKVVIPKKEEA
PEVK3:
*PPAKVPEVPKKPVPEEKVPVPVPKKVEAPPAKVPEVPKKPVPEKKVPVPAPKKVEA*
PPPKVPKKREPVPVPVALPQEEEVLFEEEIVPEEEVLPEEEEVLPEEEEVLPEEEEVLPEEEEIPPEEEEVPPEEEYVPEEEEFVPEEEVLPEVKPKVPVPAPVPEIKKKVTEKKVVIPKKEEA
*PPAKVPEVPKKVEEKRIILPKEEEVLPV*


### 4.1. Protein Production and Purification

Expression clones were transformed into chemically competent *E. coli* BL21(DE3) cells and grown overnight. The cultures were inoculated with 2% of the overnight growths, induced with 1 mM IPTG, and grown for 16 h at 30 °C while being shaken at 250 rpm. The secretion cultures were harvested by centrifugation, and the media was separated. The protein purification was initiated by either ammonium sulfate precipitation in bulk or concentrating the media down to 100–150 mL, followed by treating with 60% ammonium sulfate. The precipitated His-tagged proteins were separated by centrifugation, resuspended in immobilized metal affinity chromatography (IMAC) wash buffer (25 mM imidazole, 250 mM NaCl, 20 mM Na_2_HPO_4_), passed through a HisTrap column from Cytiva Life Sciences (Marlborough, MA, USA) and eluted using IMAC elution buffer (250 mM imidazole, 250 mM NaCl, 20 mM Na_2_HPO_4_). The purified proteins were buffer exchanged into IMAC wash buffer and added with 5% *w*/*w* TEV protease and incubated for 72 h at 4 °C to cleave the secretion tag and the HIS-tag. The TEV cleaved proteins were re-run through an IMAC column, and the proteins with the HIS-tag cleaved were collected in the wash. The protein purity was detected by SDS-PAGE, and quantification was done by Bradford assay.

### 4.2. Circular Dichroism Measurements

The proteins were diluted to a final concentration of 8 μM in either phosphate buffer (20 mM potassium phosphate buffer, pH 2–8) or HEPES buffer (20 mM HEPES, either NaCl, KCl, CaCl_2_, or MgCl_2_ pH 7.5). A concentration range of 0 to 1 M was used for NaCl and KCl and 0 to 50 mM (1.3 to 10 pCa or pMg) for CaCl_2_ and MgCl_2_. The CaCl_2_ or MgCl_2_ free samples were considered as pCa or pMg 10 and treated with 1 mM EGTA to eliminate any residual divalent metal ions. Samples were incubated at room temperature for an hour, and the spectra were measured using a JASCO J-1500 Circular Dichroism spectropolarimeter (Easton, MD, USA) using a 1 nm bandwidth and a data pitch of 1 nm using a quartz cuvette with a 1 mm pathlength at 25 °C. All the spectra were converted to molar ellipticity after background subtraction. Sample pH was verified for each sample using a micro pH probe.

### 4.3. Size Exclusion Chromatography

Purified protein samples were separated at different pHs and either CaCl_2_ or KCl concentrations using a TOSOH (Tokyo, Japan) TSKgel G3000s column on a Waters e2796 Separation Module equipped with a Waters 2489 UV/Visible Detector (Milford, MA, USA). The pH sensitivity of the PEVK proteins was tested using a phosphate buffer (20 mM potassium phosphate, 150 mM KCl) at a pH range of 3–8. Ionic sensitivity was tested using a phosphate buffer for the KCl studies (20 mM sodium phosphate, 0–1 M mM KCl) or HEPES buffer for the CaCl_2_ studies (20 mM HEPES, 1.3–10 pCa). The column was equilibrated with appropriate buffer before analysis of each set of samples, and each sample was analyzed in triplicates. The retention time was determined as the maximum absorbance at 214 nm.

## 5. Conclusions

The results of this study demonstrate that the behavior of the poly-E motif changes in the presence of PPAK sequences. The poly-E protein (PEVK2) exhibits pH-dependent fluctuations that are reduced in the presence of PPAK motifs. In addition, isolated poly-E motif protein exhibits calcium-dependent conformational shifts that are inhibited in the presence of PPAK sequences. These observations support our model that the poly-E motif sensitivity to ionic and pH fluctuations is altered by the presence of PPAK motifs. If poly-E regions have different flexibilities depending on pH and ionic strengths that are altered by PPAK regions, the elasticity of the region might be modulated by varying the combination of poly-E and PPAK motifs within the PEVK region.

## Figures and Tables

**Figure 1 ijms-23-04779-f001:**
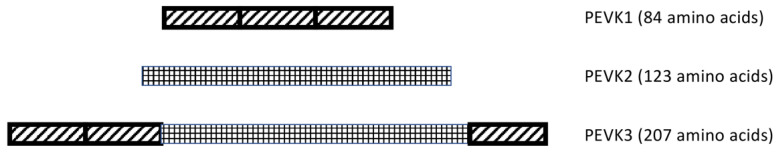
Schematic of the three constructs developed for this study.

**Figure 2 ijms-23-04779-f002:**
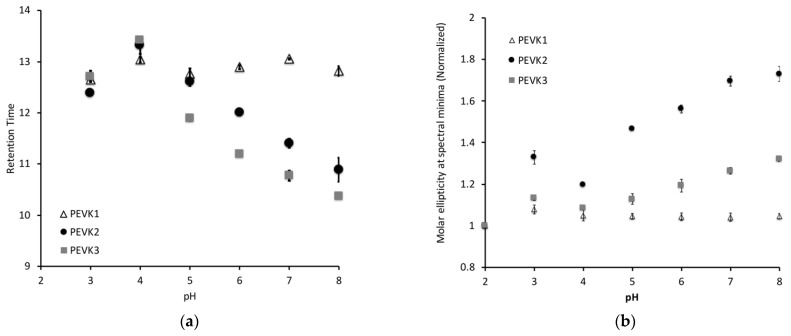
Constructs containing poly-E motif sequences exhibit pH-dependent conformational shifts. (**a**) Constructs were separated on a TOSOH TSKgel G3000s sizing column at various pHs. No retention time shifts were observed with the PPAK-containing protein (PEVK1) while both proteins containing poly-E motif sequences (PEVK2 and PEVK3) increase in retention time as pH decreases, consistent with the protein assuming a more flexible conformation. (**b**) The Circular Dichroism (CD) signal at 200 nm was measured at a range of pHs. The normalized signal shifts as a function of pH for the PEVK2 and PEVK3 constructs, consistent with the hypothesis that the glutamate content in these peptides modulates conformational flexibility. Each measurement was conducted in triplicate and error bars are included to show the standard deviation of each measurement.

**Figure 3 ijms-23-04779-f003:**
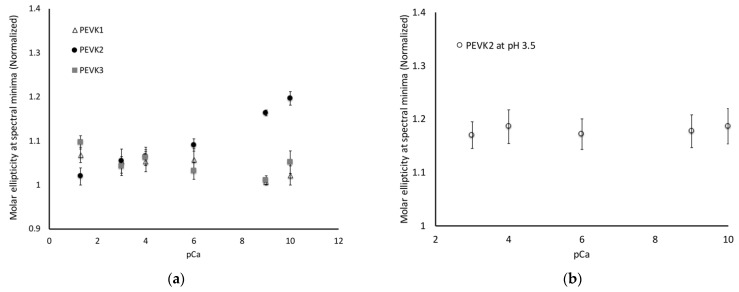
Increasing calcium concentrations induce conformational flexibility in PEVK2 construct but not PEVK3 construct. (**a**) CD signal at 200 nm was measured for all three constructs over a range of calcium concentrations. The PEVK2 construct exhibits a conformational shift that is consistent with the shift observed with shifts in pH. A similar shift is not observed in the PEVK3 construct, suggesting that the PPAK motifs alter calcium binding; (**b**) CD signal at 200 nm was measured at pH 3.5, where the glutamate motifs are protonated, and no conformational change is observed.

**Figure 4 ijms-23-04779-f004:**
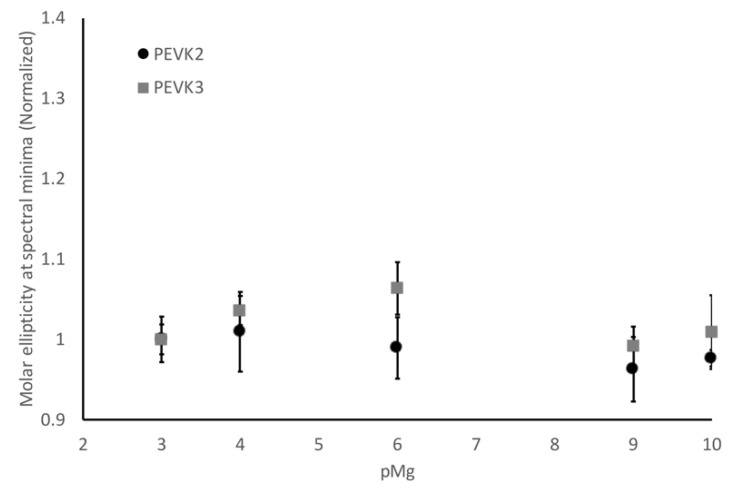
PEVK2 and PEVK3 constructs do not exhibit conformational fluctuations in the presence of Mg2+. CD signal at 200 nm was measured in the presence of increasing magnesium. No significant conformational fluctuations were observed over the concentration range tested.

**Figure 5 ijms-23-04779-f005:**
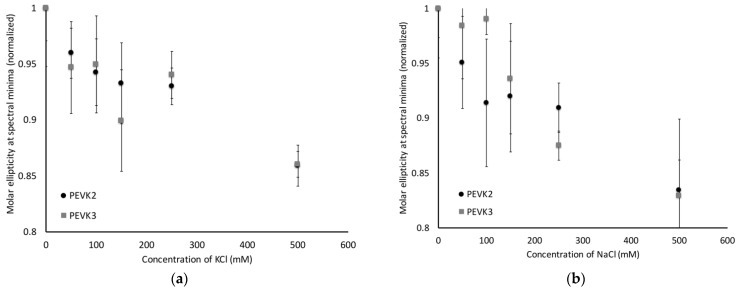
Na^+^ and K^+^ induce increased conformational flexibility at higher ionic strengths. CD signal was measured at 200 nm in the presence of increasing concentrations of K^+^ (**a**) or Na^+^ (**b**). Both constructs exhibit a decrease in the CD signal as ionic strength increases, consistent with the constructs assuming a more flexible conformation at higher ionic strengths. This supports our hypothesis that increased ionic strength will induce electrostatic shielding of the glutamates, reducing charge repulsion.

## Data Availability

All raw data is available upon request.

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
