# Peer review of "Regulation of Poly-E Motif Flexibility by pH, Ca2+ and the PPAK Motif"

_ijms, 2022, doi:10.3390/ijms23094779_

Round 1

Reviewer 1 Report

1. This manuscript demonstrates the effect of pH on the polyE motif of Titan and expands their previous work further. They have also looked at calcium dependence on these titin poly E sequences.

2. This work deals with a big elastin protein that requires extensive mechanistic and functional study. The importance of PEVK region is well known for providing elasticity. It is interesting to see the pH sensitivity of these peptides and structural changes upon interaction with Ca, which wasn’t observed for other divalent. They have shown pH dependence with shorter peptides in their 2020 paper. Taking it further to larger peptides is insignificant, however, I feel it has only added a bit more to what is already said in the 2020 paper from the group.

3. I feel adding a few more experiments like the impact of pH on fluorescence /intrinsic fluorescence, SEC-MALS will strengthen the results more.

4. Manuscript is well written with an appropriate introduction and references.

Author Response

We appreciate the reviewer recommendations.  The reviewer suggested inclusion of intrinsic fluorescence studies, but this would require development of new clones and repeating all the work completed so far, which is not really feasible.  While SEC-MALS experiments would be nice, we do not currently have access to an SEC-MALS to conduct those experiments

Reviewer 2 Report

Sudarshi et al. has presented a research work on the regulation of poly-E motif flexibility. The authors have focused on the how the pH, Ca+2 ion and the PPAK motif controls the dynamic behaviour of poly-E motif. Overall, the study is of great interest and for further improvement of the article I have few suggestions or comments to the authors.

Comments:

  • Introduction section line 35, It will be clearer if the PEVK regions can be shown with figure. Provide figure for all the construct PEVK1,2 and 3.
  • Line 96 PEVK1 construct has three PPAK motif. The line 94 states PEVK3 has two PPAK motif at N-terminal and one at C-terminal of poly-E motif, it means to generate PEVK1  poly-E motif was deleted.
  • Line 103-104 states the potassium phosphate buffer was used for experiment.The buffer pH range is pH 5.8 to 8.0 and the experiment was done in the pH range of 3 to 8
  • Figure 1: indicate the meaning of the bars on each symbol.
  • Figure 1: why the retention time and the CD spectra shows a opposite change at pH4.
  • Figure 1: line 113-114 retention time of more flexible protein is more provide reference for this sentence
  • Figure 1: SEC profile PEVK1 no change, PEVK2 and PEVK3 similar change.CD spectra PEVK3 is comparable to PEVK1.Please explain why the retention time and stoke radius is same for PEVK2 and PEVK3, when the conformational change for PEVK2 is significant
  • Over all it will be good if the number of amino acid residues are given for PPAK and poly-E motif. The number of residues will explain the impact of poly-E motif and PPAK motif on each constructs and it will be easy to understand.
  •  Line 156 and 157 change sentences to pH 6 and 8, pH 3 and 4
  • Line 145 states no impact on the hydrodynamic radius of PEVK3 but the Figure 1 shows significant change in hydrodynamic radius with pH
  • Line 146 and 147 retention times of the PEVK2 and PEVK3 constructs at pH 3 and 4 is same but figure 1 shows all the three constructs have same retention time

Author Response

We would like to thank Reviewer #2 for their thoughtful review of our work.  We have made the following changes in response to the reviewer’s comments:

  1. We have included a figure that shows the structure of the constructs used in this project, along with the number of amino acids in each construct.

  1. We adjusted the language in line 96 to reflect what the reviewer pointed out about what was deleted from the construct.

  1. The reviewer pointed out that the experiments were conducted from pH 3 to 8, which is why we used phosphate buffer. This is a polyprotic acid with multiple pKa so this allows us to cover the majority of this range.  We have added a line to the manuscript to clarify that we took pH readings before and after each experiment to verify the pH of each sample.

  1. We have added a line to the figure legend to indicate that the bars on each symbol are error bars.

  1. We have added language to clarify why the SEC and CD results show different effects with PEVK3.

  1. We have clarified the language in lanes 156 and 157 of the original draft.

  1. The impact referenced by the reviewer for line 145 of the original manuscript was observed in our previous manuscript and we are comparing our results to that study.

  1. We have adjusted the language to reflect that all three constructs have the same retention time at lower pH.

Round 2

Reviewer 1 Report

All questions are answer appropriately and reads good for publication